# The Impact of Early Acupuncture on Bell’s Palsy Recurrence: Real-World Evidence from Korea

**DOI:** 10.3390/healthcare11243143

**Published:** 2023-12-11

**Authors:** Yujin Choi, Suji Lee, Changsop Yang, Eunkyoung Ahn

**Affiliations:** 1KM Science Research Division, Korea Institute of Oriental Medicine, Daejeon 34054, Republic of Korea; choiyujin@kiom.re.kr (Y.C.); yangunja@kiom.re.kr (C.Y.); 2Department of Acupuncture and Moxibustion, Kyung Hee University Medical Center, Seoul 02447, Republic of Korea; sjstarry41@naver.com; 3KM Data Division, Korea Institute of Oriental Medicine, Daejeon 34054, Republic of Korea

**Keywords:** acupuncture, Bell’s palsy, cohort studies, facial paralysis, national health insurance data

## Abstract

Evidence-based treatment for Bell’s palsy includes the administration of steroids within 3 days of symptom onset. Additionally, a few studies have suggested the importance of combining early acupuncture treatment in the acute phase of Bell’s palsy with steroids. This study aimed to observe the impact of early acupuncture for Bell’s palsy using real-world health insurance data in Korea. This retrospective study extracted data from 45,986 adult patients with Bell’s palsy who received steroids between 2015 and 2017 with a follow-up period of at least 3 years until 2020 from the Korea National Health Insurance database. They were divided into the early acupuncture group (*n* = 28,267) and the comparison group (*n* = 17,719) based on the presence of an acupuncture treatment code within 7 days of diagnosis. The impact of early acupuncture on the likelihood of Bell’s palsy recurrence was evaluated using multivariate logistic regression. The patients in the early acupuncture group had a lower likelihood of recurrence (odds ratio: 0.81, 95% confidence interval: 0.69–0.95). This study observed a beneficial impact of early acupuncture on Bell’s palsy using real-world health insurance data in Korea. Further research is required to confirm these findings.

## 1. Introduction

Bell’s palsy refers to idiopathic facial paralysis, which presents as acute unilateral facial palsy [1]. Bell’s palsy is a common disease, with an annual incidence of 23.0 to 30.8 per 100,000 persons/year in the Korean population, estimated by the National Health Insurance [2]. The annual incidence of Bell’s palsy was reported to be 20.2 to 53.3 per 100,000 persons/year in the UK and Italian population [3,4]. The symptoms of Bell’s palsy include difficulty in closing eyelids, limited retraction of the mouth, dysfunction in forehead movement, and facial asymmetry [5,6]. Although approximately 70% of patients with Bell’s palsy are known to recover completely even without treatment [7], there are also some cases that do not recover fully and develop sequelae [1,6,8]. The recurrence of Bell’s palsy is uncommon [6], but the recurrence rate was reported to range from 0.8 to 19.4% [9]. The occurrence of Bell’s palsy and incomplete recovery likely decreases the quality of life [10] and leads to psychological problems [11]. Appropriate and effective interventions are required to shorten the time to recovery and prevent sequelae and recurrence.

The evidence-based treatment for Bell’s palsy is prescribing oral steroids within 72 h of symptom onset [1]. It is highly recommended based on the results of randomized controlled trials (RCTs) [8,12] and systematic reviews [13]. Furthermore, in East Asian countries, acupuncture has been widely used for Bell’s palsy. In China, Bell’s palsy was the most common condition in two prominent acupuncture clinics in Beijing, accounting for 20.6% and 25.3% of total outpatients [14]. In Korea, 23.3% of patients with peripheral facial palsy received Western medicine treatment only, and others also received Korean traditional medicine (KM) treatment. Acupuncture was the most frequently used KM treatment [15]. An integrative treatment package for Bell’s palsy provided in a university hospital in Korea includes corticosteroids within 72 h of symptom onset, acupuncture, and herbal medicine to suppress inflammation and nerve degeneration [16]. Despite the common use of acupuncture for Bell’s palsy in East Asian countries, it is insufficient to make an evidence-based recommendation [1,17]. In a systematic review including 40 RCTs, acupuncture was associated with a higher effective response rate for Bell’s palsy. However, the low quality and heterogeneity of the included RCTs necessitates caution in interpreting results, and further rigorous research is warranted [18].

A few studies have suggested that early acupuncture intervention is important, the same as administering steroids within 72 h in acupuncture treatment for Bell’s palsy. Patients who were administered acupuncture treatment within 7 days from the onset of Bell’s palsy showed better clinical outcomes; the time to recovery was shortened, and the occurrence of sequelae was reduced [19]. In a secondary analysis of a randomized controlled trial, patients who received acupuncture treatment in their recovery phase (21–70 days after symptom onset) of Bell’s palsy showed an increased likelihood of incomplete recovery compared with those who received it in their acute phase (1–7 days after symptom onset) [20]. In a meta-analysis, acupuncture and moxibustion for Bell’s palsy in the acute phase improved the cure rate, shortened the time to cure, and reduced the occurrence of sequelae compared with those in the non-acute phase [21].

Owing to the disease characteristics of Bell’s palsy, it is difficult to conduct a well-designed RCT of acupuncture for patients with Bell’s palsy in the acute phase. A previous RCT reported that most patients with Bell’s palsy did not want to be randomized to the treatments, and requested additional treatment if they did not experience an improvement in the early stage [22]. Further, the results of RCTs have some limitations to being generalized, because there are inclusion and exclusion criteria for enrolling patients in the trial. Therefore, this study aimed to observe the effect of early acupuncture on Bell’s palsy in the general population using real-world health insurance data in Korea.

## 2. Materials and Methods

### 2.1. Data Source and Ethics Statement

We used data from the Health Insurance Review and Assessment (HIRA) for health and medical care research. Access to this data was obtained through an application to the Healthcare Bigdata Hub (https://opendata.hira.or.kr, accessed on 14 February 2022). The research plan was reviewed and approved by the Public Data Provision Deliberation Committee of HIRA, and data suitable for the research purpose were provided through the remote analysis system (assign number: M20210730413). The ethical approval of this study was reviewed and exempted by the Institutional Review Board of the Korea Institute of Oriental Medicine (I-2105/004-004). Patient consent was not required due to the retrospective use of anonymous clinical data.

### 2.2. Study Design

This retrospective cohort study analyzed patients with Bell’s palsy extracted from claim data from the HIRA from 1 January 2015 to 31 December 2017, with a follow-up period of at least 3 years until 31 December 2020. The recurrence rate of Bell’s palsy was compared in patients with and without acupuncture treatment in the acute phase to explore the impact of acupuncture treatment in the acute phase of Bell’s palsy. Acupuncture treatment in the acute phase was defined as acupuncture treatment provided within 7 days of diagnosis. The demographic and clinical characteristics were extracted and used for analysis to adjust for covariates in the two groups (Figure 1).

### 2.3. Definition of Study Population

The study population was defined according to previous research [2,23]. The patients with Bell’s palsy were defined as meeting the following two criteria: (1) the first four digits of the main diagnostic code are “G510,” indicating Bell’s palsy, and (2) steroids are prescribed within 2 days of the first diagnosis. In the research, only patients diagnosed with Bell’s palsy were included, while patients with Ramsay Hunt syndrome or traumatic facial nerve palsy (G530 or B022) were excluded. Additionally, patients with additional diagnostic codes of facial myokymia (G514), injury of the facial nerve (S045), zoster with other nervous system involvement (B022), or fracture of the base of the skull (S021) were also excluded from the study. In addition, we included only adult patients aged 19 years or older in the analysis because pediatric patients are generally not treated with acupuncture due to concerns about pain. All types of hospital visits (ER visit, inpatient/outpatient, tertiary or secondary, primary hospitals) with Bell’s palsy as the main diagnosis were included. Patients who visited the hospital with the same diagnosis during the 1-year wash-out period were not included in this study in order to select only new-incident users with Bell’s palsy.

### 2.4. Definition of Intervention

The target intervention of this study was acupuncture in the acute phase of Bell’s palsy, defined as within 7 days of onset [19,21]. This approach is supported by two clinical practice guidelines for acupuncture treatment of Bell’s palsy developed in China and Korea [24,25]. The patients who received acupuncture treatment during the acute phase were identified as those who had acupuncture prescription codes within 7 days of their first diagnosis, and they were allocated as the early acupuncture group. To facilitate a clear comparison of the clinical effects of early acupuncture, the intervention group consisted of cases where acupuncture was performed more than twice, regardless of the type, during the acute phase. Acupuncture treatment codes included basic acupuncture, specialized acupuncture, and electric acupuncture stimulation [15]. The patients who did not have acupuncture prescription codes within 7 days of their first diagnosis were allocated to the comparison group. In addition to the timing of acupuncture treatment, both groups commonly received a range of traditional Korean medical treatments. These included moxibustion, cupping, Korean physical therapy, and herbal medicine prescriptions. These therapies were provided as complementary to the primary acupuncture treatment, reflecting the holistic approach of traditional Korean medicine in managing Bell’s palsy.

### 2.5. Definition of Outcome

Due to the limitations of using health insurance data, it was difficult to obtain information about the progress and prognosis of the disease. Therefore, the recurrence of Bell’s palsy was selected as the outcome of the study. The recurrence of Bell’s palsy could be clearly defined using the claim data. In the case of a “G510” diagnosis code and steroid prescription that occurred after a window period of >90 days after the first diagnosis, it was identified as a recurrence of Bell’s palsy [2]. This is because it is recommended to prescribe steroids for a short period of 10–14 days during the acute phase of Bell’s palsy [8,12,24]. If a “G510” diagnostic code and a steroid prescription occur simultaneously, it can be considered a recurrence. Only recurrences that occurred within the follow-up period, ranging from 3 to 6 years, could be detected. The time to recurrence, calculated only for patients who had a recurrence, was defined as the number of days from the first day of the first episode to the first day of recurrence.

### 2.6. Extraction of Demographic and Clinical Characteristics

This study collected demographic data, such as age and sex and used them as baseline variables. The onset season was divided into three categories: warm (July to October), transitional (April to June), and cold (November to March) seasons, based on the month of the diagnosis date and reflecting the Korean climate. Major comorbidities, including diabetes mellitus, hypertension, dyslipidemia, cardiovascular disease, and cerebrovascular disease, were also considered. Major comorbidities, which are underlying systemic diseases in patients with Bell’s palsy, were selected based on previous reports [23,26,27,28]. Patients who had a corresponding diagnosis code during the period from the first data collection date to the diagnosis date were defined as having a comorbidity. The diagnostic codes for each comorbidity used for data extraction based on the first three digits are as follows: diabetes mellitus (E10–E14), hypertension (I10–I13, I15), dyslipidemia (E78), cardiovascular disease (I20–I25, I50, I42), and cerebrovascular disease (I60–I69).

The treatment duration was defined as continuous if it was received within 90 days from the onset date. When the duration between the end date of the previous treatment and the start date of the next treatment was <90 days, these two visits were considered as one treatment episode. This rule also applied to multiple visits during the observation period, and the discharge date of the last visit was recorded as the end date of the treatment episode. The treatment duration was divided into two categories based on the cut-off of 90 days. As Bell’s palsy is known to subside within 3 months in most patients [29], this parameter was used to indirectly reflect the severity of Bell’s palsy in the claim data.

### 2.7. Statistical Analyses

Continuous variables are presented as mean and standard deviation (SD), while categorical variables are expressed as frequency and percentage (%). Comparative analyses of demographic and clinical characteristics were conducted between the early acupuncture and comparison groups using the independent *t*-test or chi-square test.

Subsequently, upon categorizing patients into recurred and non-recurred groups, demographic and clinical characteristics were compared between the two groups using the independent *t*-test or chi-square test. To evaluate the impact of early acupuncture treatment in the acute phase of Bell’s palsy on recurrence, a multivariable logistic regression analysis was performed. Adjusted odds ratios (ORs) and 95% confidence intervals (CIs) were calculated, accounting for potential confounding factors such as sex, age, clinical characteristics (e.g., onset season, duration of treatment, and whether early acupuncture treatment was administered), and comorbid conditions. All data preprocessing and analyses were conducted using SAS Enterprise Guide for Windows (SAS Institute Inc., Cary, NC, USA). A *p*-value of <0.05 was considered statistically significant.

## 3. Results

### 3.1. Schematic Flow

Among the 193,460 patients with G510 during 2015 to 2017, 45,986 adult patients were extracted as those with Bell’s palsy who received treatment with steroids. This number is consistent with previous reports using Korean national health insurance data, which estimated annual cases to be around 15,000 [2]. of the 45,986 patients. A total of 28,267 received acupuncture treatment within 7 days of their first diagnosis and were assigned to the early acupuncture group, while the remaining patients were assigned to the comparison group (Figure 2).

### 3.2. Demographic and Clinical Characteristics

The demographic and clinical characteristics of the included patients, based on their group (early acupuncture or comparison), are presented in Table 1. There were differences in the demographic and clinical characteristics between the two groups. The mean age was older, and there were more women in the early acupuncture group than in the comparison group. Additionally, the length of treatment was longer in the early acupuncture group, while the onset season was similar between the two groups. There were differences in baseline comorbidities, with fewer patients having diabetes mellitus but more patients having hypertension and dyslipidemia in the early acupuncture group. The overall recurrence rate observed in the study period was 1.8%, with a lower rate and a longer time to recurrence in the early acupuncture group than in the comparison group.

### 3.3. Factors Associated with Recurrence of Bell’s Palsy

The baseline demographic and clinical characteristics between the recurred and non-recurred group is presented in Table 2. The results of the multivariate logistic regression analysis can be found in Table 3. The results showed that younger age was associated with a higher likelihood of recurrence, whereas sex was not found to be associated with the likelihood of recurrence. A length of treatment exceeding 90 days was significantly associated with a higher tendency for recurrence (OR: 1.61, 95% CI: 1.32 to 1.95). In both univariate and multivariate logistic regression analyses, the early acupuncture group showed a lower likelihood of recurrence compared to the comparison group (OR: 0.81, 95% CI: 0.69–0.95). No association was observed between the presence of comorbidities and recurrence, except for hyperlipidemia. Patients with comorbid hyperlipidemia were associated with a higher risk of recurrence (OR: 1.28, 95% CI: 1.08–1.51). See Figure 3.

## 4. Discussion

The present study aimed to examine the impact of acupuncture treatment in the acute phase of Bell’s palsy using real-world health insurance data from Korea. The results showed that patients with Bell’s palsy who received early acupuncture treatment had a lower likelihood of recurrence compared to those who did not, with a significant difference after adjusting for other covariates. Although the recurrence of Bell’s palsy is uncommon [9,30], and the outcome of patients with recurrent Bell’s palsy is known to be worse compared to that of primary Bell’s palsy [31,32,33], preventing recurrence is important. Our study is the first to explore early acupuncture’s role in reducing Bell’s palsy recurrence, complementing previous research studies [19,20] that showed its benefits in accelerating recovery and enhancing complete recovery rates, using real-world health insurance data.

In this study, the recurrence rate of Bell’s palsy was 1.8%, lower than the previously reported in a systematic review and meta-analysis range of 0.8% to 19.4% with an average of 6.5% [9]. It is important to note that the follow-up period in this study was limited to a minimum of 3 years to a maximum of 6 years; hence, only recurrences that occurred within this period could be detected. Additionally, the limitations of using health insurance data should be considered when interpreting the results of this study. Owing to the nature of health insurance data, which focuses on claims, we were unable to extract important outcomes of Bell’s palsy, such as the complete recovery rate and the time to recovery. Furthermore, this was not a prospective randomized trial, and the impact of unknown confounding factors cannot be excluded. Despite these limitations, the use of health insurance data in this study provides valuable insights into the real-world impact of early acupuncture on the recurrence of Bell’s palsy in the general population of Korea.

Other factors that may influence the recurrence of Bell’s palsy were also analyzed in this study. First was the age; the results indicated that younger adults were at a higher risk of recurrence. The mean age at onset of Bell’s palsy among 140 patients with recurrent Bell’s palsy was 33 years [30]. Moreover, the incidence of Bell’s palsy is known to reach a peak between the ages of 15–45 years and is less common in those aged >60 years [29]. Second was the length of treatment exceeding 90 days, which may reflect the severity of Bell’s palsy. In this study, patients who received treatment for >90 days during their first episode of Bell’s palsy had an increased risk of recurrence compared to those who completed treatment within 90 days during their first episode. Third was the presence of systemic comorbidities. There have been reports linking recurrent Bell’s palsy with diabetes [30], although other studies have not found a correlation [31,32]. This study observed that only comorbid dyslipidemia was associated with an increased risk of recurrence of Bell’s palsy, and not diabetes or hypertension. Although a previous study using Korean national sample cohort data did not identify dyslipidemia as a risk factor for Bell’s palsy [23], another study observed a higher incidence of dyslipidemia in patients with Bell’s palsy compared to the control group [34].

Our study results showed that the length of treatment was much longer in the early acupuncture group than in the comparison group. This appears to contradict previous findings suggesting that early acupuncture can reduce the time to complete recovery in Bell’s palsy patients [19,21]. However, it is crucial to distinguish between ‘length of treatment’ and ‘time to complete recovery,’ which might explain this discrepancy. The health insurance data utilized in our study did not provide information on the exact timepoint of complete recovery. Furthermore, integrative treatment for Bell’s palsy, which includes acupuncture, often requires a longer treatment period in the clinical practice [15]. In one retrospective study, patients receiving integrative treatment, which included acupuncture, had an average treatment period of 91.9 days, while patients receiving conventional medical treatment only had an average treatment period of 21.2 days [35]. Treatment for Bell’s palsy typically ends when facial function sufficiently recovers for daily activities, yet some patients may still face discomfort and anxiety from residual symptoms such as minor asymmetry or unnatural blinking. In such instances, acupuncture treatment may be prolonged until these patient-specific symptoms are completely alleviated [36,37], resulting in an extended treatment period.

This study had limitations. First, due to the absence of severity measurement data in the National Health Insurance database, the study was unable to obtain information on the baseline severity of Bell’s palsy (initial House–Brackmann grade or electromyography results) [28], a crucial factor in determining the prognosis. Second, important outcomes of Bell’s palsy, such as complete recovery rate and time to recovery, could not be extracted from the claim data. Previous research on the effects of early acupuncture, including an observational study indicating a shorter time to recovery and reduced sequelae occurrence [19], a clinical trial focusing on the complete recovery rate [20], and a systematic review demonstrating a shorter time to recovery and a lower incidence of sequelae [21], has reported the impact of early acupuncture on these outcomes. In our study, we leveraged the advantage of claim data to observe the impact of early acupuncture on reducing the recurrence rate over an extended period. Third, this study was retrospective and therefore could not exclude potential confounding factors in examining the impact of early acupuncture on the recurrence of Bell’s palsy.

## 5. Conclusions

This study observed the impact of early acupuncture on Bell’s palsy using real-world health insurance data in Korea. The results showed that patients who received early acupuncture had a lower likelihood of recurrence than those who did not. Our study’s findings indicate that early acupuncture treatment may positively influence the prevention of Bell’s palsy recurrence. However, to generalize these results, further research incorporating initial severity information of the condition is warranted.

## Figures and Tables

**Figure 1 healthcare-11-03143-f001:**
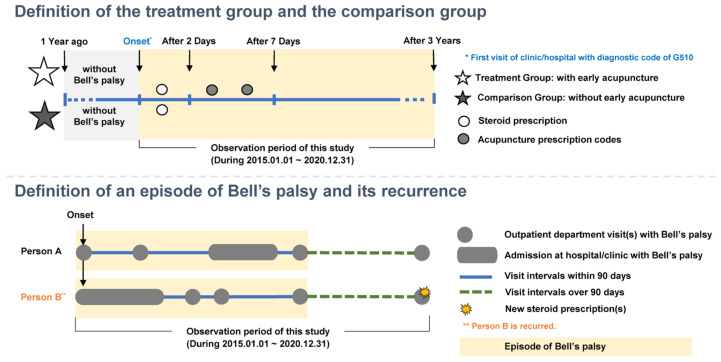
Study design.

**Figure 2 healthcare-11-03143-f002:**
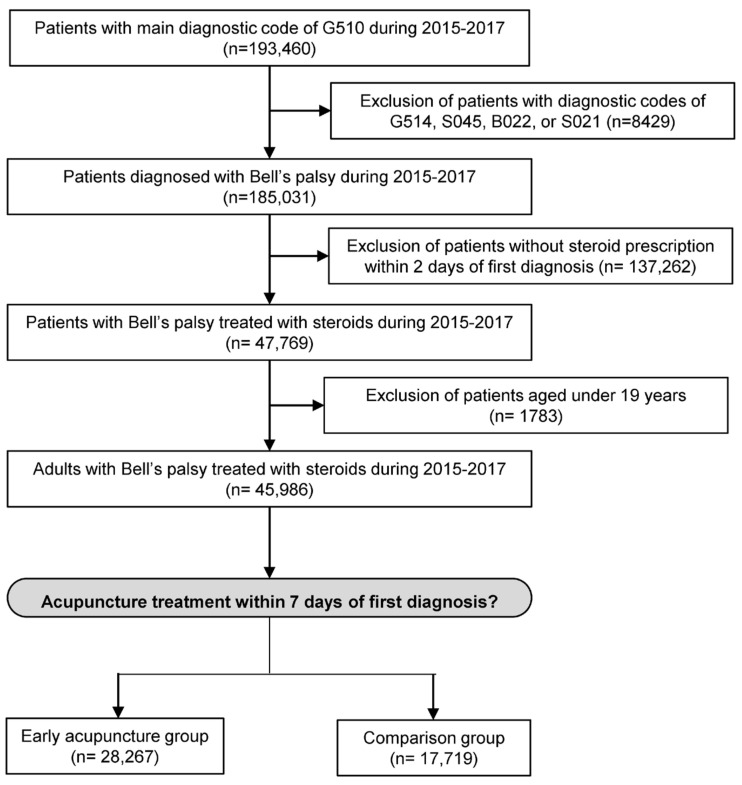
Schematic flow chart.

**Figure 3 healthcare-11-03143-f003:**
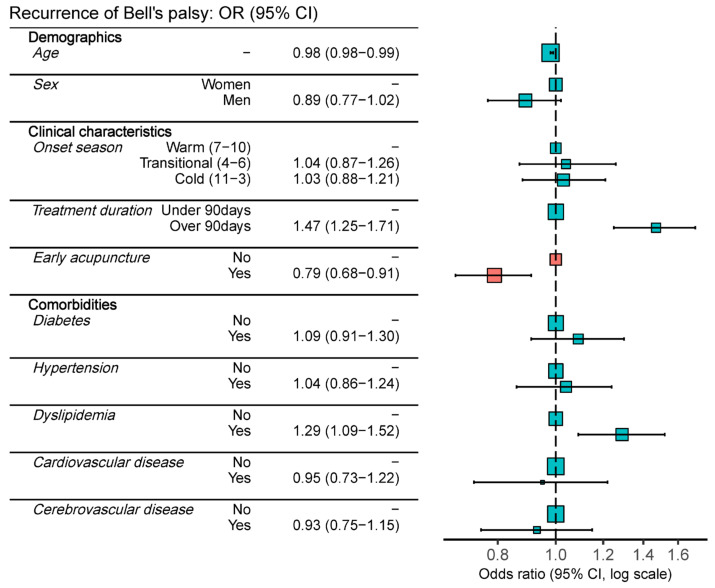
Forest plot of the adjusted odds ratios on the recurrence of Bell’s palsy. Red indicates the target intervention (early acupuncture treatment), while blue represents potential confounding factors.

**Table 1 healthcare-11-03143-t001:** Baseline characteristics of study population in two groups.

		Total (*n* = 45,986)	Early Acupuncture (*n* = 28,267)	Comparison (*n* = 17,719)	*p* Value
**Demographics**					
Age (years)		53.8 (15.7)	54.7 (15.2)	52.3 (16.2)	<0.0001
Sex	Women	22,949 (49.9%)	14,370 (50.8%)	8579 (48.4%)	<0.0001
	Men	23,037 (50.1%)	13,897 (49.2%)	9140 (51.6%)	
**Clinical characteristics**					
Onset season	Warm (7–10)	15,312 (33.3%)	9385 (33.2%)	5927 (33.5%)	0.6935
	Transitional (4–6)	10,948 (23.8%)	6766 (23.9%)	4182 (23.6%)	
	Cold (11–3)	19,726 (42.9%)	12,116 (42.9%)	7610 (42.9%)	
Treatment duration	Under 90 days	34,596 (75.2%)	19,708 (69.7%)	14,888 (84.0%)	<0.0001
	Over 90 days	11,390 (24.8%)	8559 (30.3%)	2831 (16.0%)	
**Comorbidities**					
Diabetes	No	32,778 (71.3%)	20,249 (71.6%)	12,529 (70.7%)	0.0330
	Yes	13,208 (28.7%)	8018 (28.4%)	5190 (29.3%)	
Hypertension	No	29,555 (64.3%)	17,991 (63.7%)	11,564 (65.3%)	0.0004
	Yes	16,431 (35.7%)	10,276 (36.3%)	6155 (34.7%)	
Dyslipidemia	No	25,558 (55.6%)	15,647 (55.4%)	9911 (55.9%)	0.2244
	Yes	20,428 (44.4%)	12,620 (44.6%)	7808 (44.1%)	
Cardiovascular diseases	No	41,401 (90.0%)	25,480 (90.1%)	15,921 (89.9%)	0.3215
	Yes	4585 (10.0%)	2787 (9.9%)	1798 (10.1%)	
Cerebrovascular diseases	No	39,232 (85.3%)	24,389 (86.3%)	14,843 (83.8%)	<0.0001
	Yes	6754 (14.7%)	3878 (13.7%)	2876 (16.2%)	
**Recurrence**					
Recurrence	No	45,168 (98.2%)	27,806 (98.4%)	17,362 (98.0%)	0.0026
	Yes	818 (1.8%)	461 (1.6%)	357 (2.0%)	
Time to recurrence (days)		774.6 (474.2)	834.5 (467.8)	697.2 (470.9)	<0.0001
**Concurrent treatment**					
Moxibustion	Yes	23,114 (50.3%)	17,089 (60.5%)	6025 (34.0%)	<0.0001
Cupping	Yes	29,771 (64.7%)	21,186 (74.9%)	8585 (48.4%)	<0.0001
Physical therapy	Yes	34,211 (74.4%)	24,554 (86.9%)	9657 (54.5%)	<0.0001

Data are presented as *n* (%) or mean (standard deviation).

**Table 2 healthcare-11-03143-t002:** Baseline demographic and clinical characteristics comparison between recurred and non-recurred groups.

		Not Recurred (*n* = 45,168)	Recurred (*n* = 818)	*p* Value
**Demographics**				
Age (years)		53.8 (15.7)	51.0 (15.0)	<0.0001
Sex	Women	22,522 (49.9%)	427 (52.2%)	0.1918
	Men	22,646 (50.1%)	391 (47.8%)	
**Clinical characteristics**				
Onset season	Warm (7–10)	15,044 (33.3%)	268 (32.8%)	0.9203
	Transitional (4–6)	10,749 (23.8%)	199 (24.3%)	
	Cold (11–3)	19,375 (42.9%)	351 (42.9%)	
Treatment duration	Under 90 days	34,007 (75.3%)	569 (69.6%)	0.0022
	Over 90 days	11,141 (24.7%)	249 (30.4%)	
Early acupuncture	No	17,342 (38.4%)	357 (43.6%)	0.0026
	Yes	27,806 (61.6%)	461 (56.4%)	
**Comorbidities**				
Diabetes	No	32,202 (71.3%)	576 (70.4%)	0.5853
	Yes	12,966 (28.7%)	242 (29.6%)	
Hypertension	No	29,017 (64.2%)	538 (65.8%)	0.3771
	Yes	16,151 (35.8%)	280 (34.2%)	
Dyslipidemia	No	25,129 (55.6%)	429 (52.4%)	0.0702
	Yes	20,039 (44.4%)	389 (47.6%)	
Cardiovascular diseases	No	40,657 (89.9%)	744 (91.0%)	0.4096
	Yes	4511 (10.0%)	74 (9.0%)	
Cerebrovascular diseases	No	38,522 (85.3%)	710 (86.6%)	0.2515
	Yes	6646 (14.7%)	108 (13.2%)	

Data are presented as *n* (%) or mean (standard deviation).

**Table 3 healthcare-11-03143-t003:** Multivariate logistic regression on the recurrence of Bell’s palsy.

		Estimate (SE)	Odds Ratio	95% CI	*p* Value
Intercept		−3.67 (0.33)	–	–	<0.0001
Age (years)		−0.02 (0.00)	0.98	0.98–0.99	<0.0001
Sex	Women	–	–	–	–
	Men	−0.14 (0.07)	0.89	0.77–1.02	0.0471
Onset season	Warm (7–10)	–	–	–	–
	Transitional (4–6)	0.02 (0.08)	1.04	0.87–1.26	0.8400
	Cold (11–3)	0.01 (0.09)	1.03	0.88–1.21	0.9044
Treatment duration	Under 90 days	–	–	–	–
	Over 90 days	0.00 (0.00)	1.47	1.25–1.71	0.0064
Early acupuncture	No	–	–	–	–
	Yes	−0.25 (0.07)	0.79	0.68–0.91	0.0004
Diabetes	No	–	–	–	–
	Yes	0.08 (0.09)	1.09	0.91–1.30	0.3969
Hypertension	No	–	–	–	–
	Yes	0.04 (0.09)	1.04	0.86–1.24	0.6889
Dyslipidemia	No	–	–	–	–
	Yes	0.24 (0.08)	1.29	1.09–1.52	0.0046
Cardiovascular diseases	No	–	–	–	–
	Yes	−0.05 (0.13)	0.95	0.73–1.22	0.6870
Cerebrovascular diseases	No	–	–	–	
	Yes	−0.07 (0.11)	0.93	0.75–1.15	0.5541

Adjusted odds ratios are presented as the results of multivariate logistic regression.

## Data Availability

Restrictions apply to the availability of these data. Data were obtained from Healthcare Bigdata Hub of Health Insurance Review and Assessment (HIRA) in Korea and are available at https://opendata.hira.or.kr (accessed on 2 February 2022) with the permission of HIRA.

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
