# Peer review of "The Impact of Early Acupuncture on Bell’s Palsy Recurrence: Real-World Evidence from Korea"

_healthcare, 2023, doi:10.3390/healthcare11243143_

Round 1

Reviewer 1 Report

Comments and Suggestions for Authors

This is an interesting look at the use of acupuncture data to try to find an effect of acupuncture. Since recurrence is an unlikely event (and even more so at 3-6 years) there is little here that impacts how acupuncture should be delivered.

As the authors suggest, better would be to understand the impact of the early acupuncture intervention on outcomes - but that is not possible with this data. 

Other items of interest may be cost. What is the cost differential to provide early acupuncture vs. not. Does the cost of this additional therapy make sense if it is only reducing recurrence some - obviously outcomes data would be more beneficial here.

Also wonder if you could see any difference if someone started acupuncture more than 7 days after the index visit would it be less effective for recurrence - or would those individuals need more treatment?

Nevertheless, it is valuable information that is relayed. Understanding the number of cases, the frequency of use of acupuncture and this impact is valuable. 

Comments:

1. Maybe could have done a 3 year wash-out period because some may be experiencing a second episode of Bell's Palsy. 

2. Were treatments other than acupuncture technique used? The introduction mentions herbal medicine, moxibustion. Maybe better describe that the intervention is a full scope of acupuncture to include acupuncture, herbal medicine and adjunctive therapies.

Author Response

Comment 1:

This is an interesting look at the use of acupuncture data to try to find an effect of acupuncture. Since recurrence is an unlikely event (and even more so at 3-6 years) there is little here that impacts how acupuncture should be delivered.

As the authors suggest, better would be to understand the impact of the early acupuncture intervention on outcomes - but that is not possible with this data. 

Other items of interest may be cost. What is the cost differential to provide early acupuncture vs. not. Does the cost of this additional therapy make sense if it is only reducing recurrence some - obviously outcomes data would be more beneficial here.

Also wonder if you could see any difference if someone started acupuncture more than 7 days after the index visit would it be less effective for recurrence - or would those individuals need more treatment?

Nevertheless, it is valuable information that is relayed. Understanding the number of cases, the frequency of use of acupuncture and this impact is valuable. 

Response 1:

Thank you for your insightful review. In addressing the impact of acupuncture on Bell's palsy recurrence, it's crucial to highlight that our study's comparison group included patients who solely received Western medical treatments. This context underscores the significance of a substantial number of Bell's palsy patients choosing acupuncture within 7 days of onset. Such a trend aligns with traditional Korean clinical practice guidelines and suggests that early acupuncture is being effectively implemented in disease management. Our study offers initial evidence that early acupuncture may play a role not only in treating the disease but also in potentially reducing recurrence rates. This is a noteworthy finding, especially considering its absence in previous large-scale studies.

Regarding cost considerations, a thorough analysis is required to estimate the financial implications of disease recurrence, particularly in the realm of treatment cost-effectiveness. This area promises to be a crucial aspect of future research.

Furthermore, the concept of conducting separate analyses on patients who underwent acupuncture, with a specific focus on the timing of the initial treatment (whether within or after 7 days of onset), opens up an intriguing research avenue. We believe such investigations could yield more profound insights into the efficacy of acupuncture in Bell's palsy management. These topics are high on our agenda for future research endeavors.

Lastly, we deeply value your perceptive feedback, which will significantly influence the direction of our future studies. Your insights are extremely valuable to our team, and we are confident they will lead to more comprehensive research. We eagerly anticipate integrating your suggestions into our ongoing and future research projects.

Comment 2:

Maybe could have done a 3-year wash-out period because some may be experiencing a second episode of Bell's Palsy. 

Response 2:

We appreciate your suggestion regarding the necessity of a 3-year wash-out period to account for potential second episodes of Bell's palsy. To explore this further, we would need to extract data for subjects with a minimum 3-year observation period. Our study operates within a system where we can only analyze data that has been pre-approved for specific periods. Extending the recurrence period to 3 years would change the entire cohort of our study subjects. We hope to conduct research with a 3-year observation period in the future.

Comment 3:

Were treatments other than acupuncture technique used? The introduction mentions herbal medicine, moxibustion. Maybe better describe that the intervention is a full scope of acupuncture to include acupuncture, herbal medicine and adjunctive therapies.

Response 3:

The traditional Korean treatments provided to Bell's palsy patients often include a combination of acupuncture, moxibustion, cupping, and physical therapy. To clarify this, we will elaborate on these treatments in the 2.4. Definition of intervention section (p.3, lines 127-132) and supplement this information in Table 1.

(p.3, lines 127-132) The presence or absence of early acupuncture was considered when classifying the two groups. Participants in both groups received interventions other than acupuncture, such as moxibustion, cupping, physical therapy, or herbal medicine prescriptions, or did not receive any additional interventions. The details of the concurrent treatments administered to individuals in each group have been separately documented (Table 1)

Thank you once again for your valuable feedback, which will undoubtedly enhance the quality and clarity of our research.

Reviewer 2 Report

Comments and Suggestions for Authors

This is a large retrospective study based on data extracted from the Korea National Health Insurance database of 45,986 patients with Bell's palsy, some of whom received an early acupuncture treatment, and were followed up for 3 years. The aim of the study was to evaluate the likelihood of the Bell's palsy recurrence in patients treated with acupuncture, as compared to the control group.  The authors report that early acupuncture group had a lower likelihood of Bell's palsy recurrence.

While the authors should be congratulated for the effort invested in performing such a large study, it however had several methodological flaws:

1.       There are significant differences in comorbidities between the two groups. Please elaborate on how these groups can be compared.

2.       Taking in to account that the Korea National Health Insurance database did not contain information on the baseline severity of the Bell's palsy, which as the authors state, is a "crucial factor for determining the prognosis", analyzing other factors such as age and length of treatment on the recurrence risk, may be of a limited importance.

3.       Table 2 actually consists of two tables: 1. clinical characteristics of the whole study population, and 2. additional grouping according to the treatment (acupuncture yes/no), and thus is difficult to comprehend.

Also, please elaborate on the Odds Ratio adjustment.

Comments on the Quality of English Language

please consider scientific editing

Author Response

Comment 1:

There are significant differences in comorbidities between the two groups. Please elaborate on how these groups can be compared.

Response 1:

Thank you for your valuable feedback and for acknowledging the scale of our study. We appreciate your constructive comments and would like to address the methodological concerns you raised (p.4, lines 172-179).

(p.4, lines 171-180) The risk of recurrence associated with each individual variable was assessed using univariate logistic regression analysis, which provided the Crude Odds Ratios (Crude ORs). To assess the impact of early acupuncture treatment in the acute phase of Bell’s palsy on recurrence, while adjusting for potential confounding factors, a multivariable logistic regression analysis was conducted. Both crude and adjusted odds ratios (OR) and 95% confidence intervals (CIs) were calculated. In the multivariable logistic regression analysis, adjustments were made for risk factors between the two groups, including gender, age, clinical characteristics (such as onset season, duration of treatment, and whether early acupuncture treatment was received), and comorbid conditions.

Comment 2:

Taking in to account that the Korea National Health Insurance database did not contain information on the baseline severity of the Bell's palsy, which as the authors state, is a "crucial factor for determining the prognosis", analyzing other factors such as age and length of treatment on the recurrence risk, may be of a limited importance.

Response 2:

You correctly pointed out the absence of baseline severity data in the Korea National Health Insurance database. We agree that this is a limitation of our study. However, we attempted to overcome this by analyzing other relevant factors such as age and length of treatment, which could provide indirect insights into the recurrence risk. While these factors may have limited importance compared to baseline severity, they still contribute valuable information to the overall understanding of Bell's palsy management. We will further clarify this point in our revised manuscript (p.9, line 282-285).

(p.9, line 289-298) Firstly, the absence of severity measurement data in the National Health Insurance database precluded the acquisition of information concerning the baseline severity of Bell's palsy, including initial House-Brackmann grade or electromyography results [28], —critical factors for prognosis determination.

Comment 3:

Table 2 actually consists of two tables: 1. clinical characteristics of the whole study population, and 2. additional grouping according to the treatment (acupuncture yes/no), and thus is difficult to comprehend.

Also, please elaborate on the Odds Ratio adjustment.

Response 3:

You correctly pointed out the absence of baseline severity data in the Korea National Health Insurance database, which we acknowledge as a limitation of our study. To address this, we analyzed other relevant factors such as age and length of treatment, providing indirect insights into the recurrence risk. While these factors may not be as crucial as baseline severity, they still offer valuable insights into the management of Bell's palsy. Additionally, in our statistical analysis, the crude odds ratio (OR) represented the risk of recurrence associated with each variable individually, while the adjusted OR, derived from multivariable logistic regression, reflected the recurrence risk considering all variables collectively. This distinction between crude and adjusted ORs is an important aspect of our analysis, and we will ensure to clarify this in our revised manuscript (p.4, lines 170-172). Additionally, we have added a description of the Crude Odds Ratios (ORs) at the bottom of Table 2 to provide a clearer understanding of how these values were derived and their significance in the context of our study.

(p.4, lines 170-172) The risk of recurrence associated with each individual variable was assessed using univariate logistic regression analysis, which provided the Crude Odds Ratios (Crude ORs). To assess the impact of early acupuncture treatment in the acute phase of Bell’s palsy on recurrence, while adjusting for potential confounding factors, a multivariable logistic regression analysis was conducted. Both crude and adjusted odds ratios (OR) and 95% confidence intervals (CIs) were calculated. In the multivariable logistic regression analysis, adjustments were made for risk factors between the two groups, including gender, age, clinical characteristics (such as onset season, duration of treatment, and whether early acupuncture treatment was received), and comorbid conditions.

Round 2

Reviewer 2 Report

Comments and Suggestions for Authors

My comment 2 referred to the possible conclusion of the limited effectiveness of acupuncture in your research. The stated lack of information (of the baseline severity of the disease, the complete recovery rate and the time to recovery, which could not be extracted from the data), significantly limits a generalizability effect.

Although your conclusion of the necessity of additional research in a way reduces the flaw in interpretation, some more references or considerations may add to the validity of the study.

Also, the mentioned Table-2 may be simplified for clarity or divided into two tables.

An additional comment may be added: please consider simplifying the text for the sake of a reader  

Comments on the Quality of English Language

Author Response

Comment 1:

My comment 2 referred to the possible conclusion of the limited effectiveness of acupuncture in your research. The stated lack of information (of the baseline severity of the disease, the complete recovery rate and the time to recovery, which could not be extracted from the data), significantly limits a generalizability effect.

Although your conclusion of the necessity of additional research in a way reduces the flaw in interpretation, some more references or considerations may add to the validity of the study.

Response 1:

Thank you for your valuable feedback. We acknowledge the limitations of our study, particularly the absence of information on the baseline severity of Bell's palsy, complete recovery rate, and time to recovery in our database. This does indeed impact the generalizability of our findings. In our conclusion, we have emphasized the need for additional research to address these limitations. To enhance the validity of our study, we have incorporated additional references as suggested.

(p.9, lines 282-290) Second, important outcomes of Bell’s palsy such as complete recovery rate and time to recovery, could not be extracted from the claim data. Previous research on the effects of early acupuncture, including an observational study indicating a shorter time to recovery and reduced sequelae occurrence [19], a clinical trial focusing on the complete recovery rate [20], and a systematic review demonstrating a shorter time to recovery and a lower incidence of sequelae [21], have reported the impact of early acupuncture on these outcomes. In our study, we leveraged the advantage of claim data to observe the impact of early acupuncture on reducing the recurrence rate over an extended period.

Comment 2:

Also, the mentioned Table-2 may be simplified for clarity or divided into two tables.

Response 2:

We appreciate your suggestion regarding the structure of Table 2. In response, we have revised the table for improved clarity by dividing it into two separate tables.

(p.6-7, lines 217-221)

Table 2. Baseline demographic and clinical characteristics comparison between recurred and non-recurred groups

Table 3. Multivariate logistic regression on the recurrence of Bell’s palsy

Comment 3:

An additional comment may be added: please consider simplifying the text for the sake of a reader.

Response 3:

Thank you for your constructive input. We have endeavored to simplify the text for better reader comprehension. These revisions aim to enhance the clarity and accessibility of our research.

(p.9, lines 259-262) Although a previous study using Korean national sample cohort data did not identify dyslipidemia as a risk factor for Bell's palsy [23], another study observed a higher incidence of dyslipidemia in patients with Bell's palsy compared to the control group [34].

(p.9, lines 264-269) This appears to contradict previous findings suggesting that early acupuncture can reduce the time to complete recovery in Bell’s palsy patients [19,21]. However, it's crucial to distinguish between 'length of treatment' and 'time to complete recovery,' which might explain this discrepancy. The health insurance data utilized in our study did not provide information on the exact timepoint of complete recovery.

(p.9, lines 296-299) Our study's findings indicate that early acupuncture treatment may positively influence the prevention of Bell's palsy recurrence. However, to generalize these results, further research incorporating initial severity information of the condition is warranted.